# Progressive Early-Onset Leukodystrophy Related to Biallelic Variants in the *KARS* Gene: The First Case Described in Latin America

**DOI:** 10.3390/genes11121437

**Published:** 2020-11-29

**Authors:** Adriana Vargas, Jorge Rojas, Ivan Aivasovsky, Sergio Vergara, Marianna Castellanos, Carolina Prieto, Luis Celis

**Affiliations:** 1Clínica Universidad de La Sabana, Km 7, Autopista Norte de Bogotá, Chía 250001, Colombia; 2Faculty of Medicine, Pontificia Universidad Javeriana, Cra 7a N° 40 B-36, Bogotá 110231, Colombia; jorgerojas.martinez@gmail.com; 3Faculty of Medicine, Universidad de La Sabana, Km 7, Autopista Norte de Bogotá, Chía 250001, Colombia; sergiovercar@unisabana.edu.co (S.V.); mariannacafe@unisabana.edu.co (M.C.); andreaprso@unisabana.edu.co (C.P.); luis.celis@unisabana.edu.co (L.C.)

**Keywords:** *KARS* gene, aminoacylation, leukodystrophy, epilepsy, hearing loss developmental delay, whole exome sequencing

## Abstract

The *KARS* gene encodes the aminoacyl-tRNA synthetase (aaRS), which activates and joins lysine with its corresponding transfer RNA (tRNA) through the ATP-dependent aminoacylation of the amino acid. *KARS* gene mutations have been linked to diverse neurologic phenotypes, such as neurosensorial hearing loss, leukodystrophy, microcephaly, developmental delay or regression, peripheral neuropathy, cardiomyopathy, the impairment of the mitochondrial respiratory chain, and hyperlactatemia, among others. This article presents the case of a Colombian pediatric patient with two pathological missense variants in a compound heterozygous state in the *KARS* gene and, in addition to the case report, the paper reviews the literature for other cases of *KARS1*-associated leukodystrophy.

## 1. Introduction

Aminoacyl t-RNA synthetases (aaRSs) are 20 enzymes that are fundamental in the translation of messenger RNA (mRNA) to proteins in eukaryotic cells. Each aaRS mediates the initial adenylation of the amino acid and the subsequent formation of an ester bond between aminoacyl-AMP and its specific t-RNA; this process is denominated aminoacylation, and it can take place in the cytoplasm, mitochondria, or in both compartments in the cell [1,2,3,4,5,6].

Lysyl-tRNA synthetase (LysRS), which is a bifunctional aaRSs, is encoded by the *KARS* gene (OMIM # 601421). This gene is located on the chromosome 16q23.1 and encompasses 15 exons; it is translated into the two Lysyl-tRNA synthetase protein (LysRS) isoforms through alternative splicing. These forms of the protein differ mainly in their amino terminal end, which defines their final location in the cytoplasm or in the mitochondria [2,7,8].

Due to the central role of aaRSs in decoding the genetic code, the alteration of their function leads to various diseases with an autosomal recessive pattern of inheritance, with a broad range of neurological phenotypes with or without systemic involvement.

Pathogenic variants in the *KARS* gene have been associated with autosomal recessive non-syndromic sensorineural hearing loss, congenital visual impairment, progressive microcephaly, leukodystrophy, some variants of Charcot–Marie–Tooth disease, severe cardiomyopathy related to mild to severe myopathy, developmental delay, cognitive impairment, and epilepsy, among others [2,4,8,9].

## 2. Materials and Methods

Before the initiation of the study, acceptance was obtained from the Academic Research Ethics Committee of the Clinica Universidad de la Sabana, and informed consent was signed by the patient’s parents under the guidelines of the Declaration of Helsinki; after this, a review of the patient’s clinical record and a physical examination of the patient were performed.

Trio Whole Exome Sequencing (trio WES) of the patient and his parents DNA was requested, including the analysis of approximately 23,000 genes. Exon capture was completed with the Nextera Exome Capture^®^ System, followed by NGS (Next Generation Sequence) sequencing, which was completed by the Illumina HiSeq 4000^®^ platform, with a median depth of 66x. Finally, the final variants in the *KARS* gene were identified following bioinformatic protocols while using the Homo sapiens (human) genome assembly *GRCh37* from the Genome Reference Consortium.

A literature search was performed in databases such as Clinicalkey, PubMed, Access Medicine, and OMIM in order to proceed with a comparison of the case with the available information.

## 3. Results

### 3.1. Clinical Case

A male patient from Colombia was first evaluated by pediatric neurology at six months of age because of a history of a global development delay and an absence of responses to auditory stimuli. He is the first child of non-consanguineous parents, with an unremarkable gestation, born at term by assisted vaginal delivery, and without any other complications (Figure 1). In the family history, the father reported a paternal aunt with seizures and motor impairment of an unknown cause, with apparently no auditive nor visual dysfunction, who died during adolescence, a fact which may suggest an autosomical recessive hereditary pattern of the genetic mutation regarding the father’s offspring.

At the age of twelve months, the patient suffered sudden neurological regression associated with an intercurrent febrile event of unclear etiology; it comprised the loss of language and all the motor milestones, and even head support. Additionally, visual deterioration was evident due to the absence of fixation and object tracking by the child.

During the periodical follow-up, the neurological decline of the patient included a progressive paresis of the four limbs and impaired swallowing, making the use of gastrostomy necessary. At 6 years old, the patient presented focal motor-tonic seizures which were treated and controlled with levetiracetam and valproic acid.

On physical examination, the patient does not have eye-gaze fixation, nor following; there is no answer to auditory stimuli, nor following of commands. The child does not emit any verbal language, and has spastic quadriplegia, without cephalic control or the manipulation of objects; fasciculations in the abdomen were also evident.

#### 3.1.1. Laboratory Studies

Studies of lactate, pyruvate, and quantitative amino acids were conducted with high-performance chromatography in blood and urine; the enzymatic activity of hexosaminidase A, ammonia, and serum long-chain fatty acids was found to be at normal values.

#### 3.1.2. Electrophysiological Studies

The auditory evoked potentials at 10 months of life showed severe sensorineural hearing loss; on the other hand, the visual evoked potentials taken at one year of age presented prolonged latencies of N75-P100-N145 waves, suggestive of the demyelination of the retino-genico-calcarin pathway.

#### 3.1.3. Images

Magnetic resonance imaging of the brain (MRI) performed at 10, 16, and 96 months old showed a marked progressive cortical atrophy associated with diffuse ventriculomegaly. Additionally, enhanced signal in the T2-weighted and FLAIR images in the splenium of the corpus callosum; thalamus; and deep white matter of the temporal, parietal, and occipital lobes and the cerebellum were evident. There are no areas of restricted diffusion in the DWI and ADC maps. The spectroscopy revealed a decreased *N*-acetyl-aspartate peak, in contrast to a high peak of lactate (Table 1).

#### 3.1.4. Genetic Studies

In the absence of an etiology of the patient’s condition, a WES of his DNA was performed with 92% coverage including all coding exons. Two variants in the *KARS* gene (OMIM *601421) were identified; the first variant NM_153460.4(IL17RC): c.1514G>A *(p.Arg505His)* was classified as pathogenic (rs778748895), as previously reported in the literature [4,12,13].

This variant is very rare in the population, with a low allelic frequency in population databases; it has an allele frequency of 0.00001 in the Exome Aggregation Consortium (ExAC) and an allele frequency of 0.00000 in The Genome Aggregation Database (gnomAD). The second variant detected was NM_001130089.1: c.1577C>T (p.Ala526Val), (rs1415687857), which has been classified as probably pathogenic and has been reported in the literature; this variant also has an extremely low allelic frequency, reported in the gnomAD as 0.00003. These variants are in a trans configuration, which confirms a state of compound heterozygosity, because they were identified in the heterozygous state in each parent of the patient.

Genes search:Clinical VAR: National Center for Biotechnology Information. ClinVar; [VCV000694746.2], https://www.ncbi.nlm.nih.gov/clinvar/variation/VCV000694746.2 (accessed 12 February 2020).Prediction In Silico: Varsome Clinical. A Clinical-grade Platform for interpretation of NGS Data, https://varsome.com/variant/hg19/NM_001130089.1(*KARS*)%3AA526V(accessed 10 April 2020).

## 4. Discussion

A variable clinical presentation of the disease has been associated with *KARS* gene mutations; all those disorders have been linked to an autosomal recessive inheritance pattern mechanism [11,12]. Among the most affected organs are those with a high energy demand, such as the brain, heart, skeletal muscle, and kidneys, probably secondary to the mitochondrial disfunction with a secondary impaired oxidative phosphorylation [19].

The wide spectrum of neurological phenotypes comprises the peripheral neuropathies with a variable severity of disease, microcephaly, developmental delay, cognitive decline, epilepsy, ataxia, hypotonia, spasticity, hemiplegia, quadriparesis, abnormal movements such as dystonia and chorea, leukoencephalopathy, cerebral white matter, brainstem and spinal cord calcification, visual impairment, and sensorineural hearing loss [2,4,9,12,13,15,16,19,20].

The leukodystrophies comprise genetic diseases that predominantly affect the white matter, with the damage of the glial cells and myelin sheaths [16,21]. One study identified gliosis and demyelination through the brain samples of two individuals who did not have significant structural changes in their cerebral cortex, despite having documented *KARS* mutations [14,15]. The pathologic and image findings shown in our patient and in the literature support the classification of the white matter injury, associated with LysRS mutations, as a primary leukodystrophy (LD) [20,22].

Imaging studies represent a gateway to the diagnostic approach, with a wide repertory of findings described in *KARS* LD. These include the progressive thinning, with symmetric hyperintensity, of the cerebral white matter in the T2 and FLAIR images, including or not the U fibers [2,13,19].

Additionally, cortical atrophy; gliotic changes; hypomyelination; demyelination; ventriculomegaly; increased signal in the internal capsule, corticospinal tracts, thalamus, substantia nigra, and cerebellar peduncles; and corpus callosum dysgenesis have been described [2,10,11,12]. In the spectroscopy, a reduced *N*-acetyl-aspartate peak with elevated or diminished lactate peaks in the white matter have been found [12,13,20,22].

A remarkable sign that has been described in some patients with the *KARS* mutation is the calcification of the cerebral white matter, basal ganglia, internal capsules, cerebellar nuclei, brainstem, and spinal cord [12,13,16].

Our patient presents a severe neurological phenotype, similar to what has been described previously in the literature, with a marked clinical deterioration over time. The brain images showed an extensive and rapidly progressive cortical atrophy, a loss of white matter, corpus callosum dysgenesis, calcifications, and a gliotic lesion of the white and gray matter in the occipital lobes.

For undiagnosed patients with white matter disorders of suspected genetic etiology, the WES has emerged as a diagnostic test that allows use to determine a definitive etiology in approximately 70% of the patients; this figure could be increased to 80% with the use of genome sequencing [23,24]. In our case, the use of WES in triplicate allowed the achievement of a definitive diagnosis for the patient.

*KARS* mutations have been repeatedly linked to non-syndromic hearing impairment without other neurological relevant symptoms; Santos-Cortez et al. demonstrated this linkage in 13 patients with isolated hearing impairment with variations in the *KARS* gene (c.1517T>C and c.1129G>A). Ref. [8] *KARS* expression has been specially demonstrated in animal models at the Organ of Corti, specifically in the spiral ligament of the cochlea, inner and outer hair cells, tectorial membrane, supportive Deiter’s cells, basilar membrane, spiral ligament, spiral limbus epithelium, and the inner sulcus cells, therefore explaining the relation between any alteration of this gene with the normal development of these structures. The *KARS* mutations have been proposed as the cause of the secondary malfunction of the structures previously indicated in the inner ear, as they are in charge of the transduction of the mechanic stimuli to an electric signal; this has been proposed but still requires further study [8]. In our patient, this could explain the early hearing loss, which could also deteriorate in time, secondary to the white matter progressive damage.

As seen, it is known what the impact is of some of the mutations in the *KARS* domain; nonetheless, there are still multiple *KARS* gene variations with unestablished clinical significance until this day, and even most of those classified as pathogenic are not supported nor related to phenotypical reports. In fact, Ardissone et al. determined that only 27 patients have a phenotype related to a *KARS* mutation [12], with this in relation to the fact that there 120 variations that involve this gene have been reported in ClinVar, of which 31 have been classified as pathogenic (including the first variation found in the patient) and four with conflicting interpretations (where we found the second variant of our patient). It can now be stated that this second variant can now be classified as pathogenic (Table 1).

The use of WES in this patient resulted in the identification of two variants in the *KARS* gene that were identified in a heterozygous state in each parent of the patient, confirming their carrier status; the first pathogenic variant has already been reported in the literature, c.1514G>A (p.Arg505His).

This variant has been identified in two cases of patients with sensorineural hearing loss and leukodystrophy [4,12,13].

This amino acid change occurs in a highly conserved position in various species, so the in silico prediction concludes that it is deleterious. This variant is present in heterozygosity in one of 138,000 individuals in the world population, and has been shown in vivo to affect enzyme activity, leading to a decrease in lysine t-RNA aminoacylation, and is therefore considered pathogenic.

Although the second variant, c.1577C>T (p.Ala526Val), has been previously described in a heterozygous state in one instance in the medical literature [16], it has not been reported to OMIM, ClinVar and Human Genome Mutation Database (HGMD) in any other affected individuals, and it has never been associated, with full certainty, with phenotypes related to *KARS* mutations in humans. Moreover, this variation found in the aforementioned patient has been classified as “Likely pathogenic”, and there is no clinical certainty that it may generate a specific phenotype; however, due to the highly conservative state of this residue, it is highly probable that a mutation on this specific site, as well as in the first mutation, could damage the catalytic domain of the LysRS, which contains the site to activate the amino acid lysine and links it to the corresponding t-RNA through aminoacylation, leading to a final reduction in the amino-acylation of l-lysine t-RNA.

These conservative changes, as seen in the second variant, can be classified as deleterious by solely analyzing them with 16 bioinformatic predictors (https://varsome.com/variant/hg19/NM_001130089.1 (*KARS*)% 3AA526V), and also by relating them to the phenotypical characteristics observed in our patient.

Mutations of the *KARS* gene represent a diagnostic challenge given the scarcity of the available literature, in addition to the wide clinical phenotype, which could be an isolated symptom, such as non-syndromic hearing loss or a mild to severe neurological disease [2,4,8,12,13,16,20,22]. In addition, identifying a *KARS* gene mutation as the etiology in a specific patient could impact management, prognosis, and genetic counseling.

## 5. Conclusions

This is the first case of a patient with the *KARS* mutation associated with a severe neurological phenotype reported in Latin America. The pathogenic variants in the *KARS* gene are responsible for the LysRS deficiency that generates various alterations in neurological development and mitochondrial function; however, new phenotypes attributable to these variants continue to emerge today as a recognizable cause of rare diseases in the pediatric population.

Given the variety of clinical expression for mutations in *KARS*, our case shows that the biallelic pathogenic variants in this gene may be responsible for the severe clinical phenotype that is highlighted by hearing loss and severe early-onset leukodystrophy, which is probably the result of a reduced activity of LysRS.

It is important in children with cognitive and neurological impairment with extensive compromise of the central nervous system white matter to promote the use of WES and genome sequencing if the initial metabolic and enzymatic assays are negative. The availability of these diagnostic tools has significantly increased the likelihood of achieving a definitive diagnosis in most of the patients with a suspected genetic etiology of the disease.

## Figures and Tables

**Figure 1 genes-11-01437-f001:**
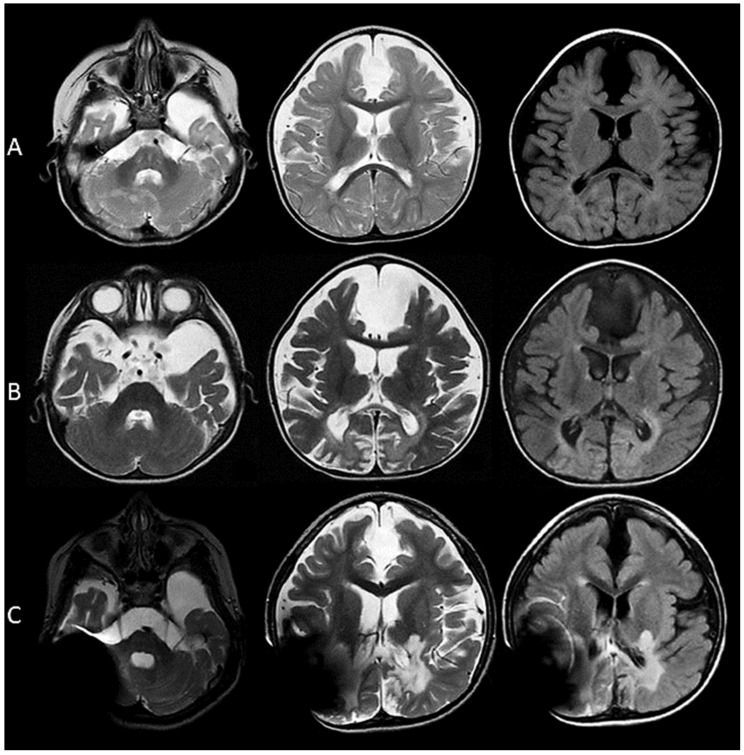
(**A**) At 10 months of age, a prominent subarachnoid space is evident, secondary to a mild degree of cortical atrophy, with a remarkable volume loss of the frontal and temporal operculum. The myelination for the age is adequate; also, interhemispheric and left temporal arachnoid cysts are seen. There is no diffusion restriction in the DWI (Diffusion-weighted Imagin) and ADC (Apparent Diffusion Coefficient) maps. (**B**) At 16 months of age, a marked generalized cortical atrophy is evident. In the T2-weighted and FLAIR images, areas of increased signal are observed predominantly in the deep white matter of the parietal and occipital lobes; also, gliotic changes are present bilaterally in the occipital poles. A compensatory ventriculomegaly is evident. There is no enhancement with gadolinium. (**C**) At 44 months of age, there is a large signal artifact which limits the observation of the right parietal, temporal, and occipital lobes, as well as the right cerebellar hemisphere. In the T2 and FLAIR images, there is an increased signal in the lateral aspect of thalamus bilaterally; the posterior arm of the internal capsules; the splenium of the corpus callosum; and again of the white matter of the temporal, parietal, and occipital lobes. Additionally, an increased signal is seen in the white matter of the left cerebellar hemisphere. A decreased volume of the brainstem is observed as well.

**Table 1 genes-11-01437-t001:** Current *KARS1* gene variations documented up until this day. Relation to their clinical significance and the available evidence.

Variation/Location	Clinical Significance	Reference
NM_001130089.1(*KARS1*): c.1129G>A (p.Asp377Asn) *GRCh37:* Chr16:75663371 *GRCh38:* Chr16:75629473	Pathogenic	Santos-Cortez et al. [8] Szafranski et al. [10] Basit S et al. [11]
NM_005548.2(*KARS1*): c.1430G>A (p.Arg477His) *GRCh37:* Chr16:75663434 (First Variant) *GRCh38:* Chr16:75629536	Pathogenic	Zhou et al. [4] Ardissone et al. [12] C Sun et al. [13] THIS PAPER (2020)
NM_001130089.1(*KARS1*): c.1438del (p.Leu480fs) *GRCh37:* Chr16:75664391 *GRCh38:* Chr16:75630493	Pathogenic	Clinical testing, NO phenotype correlation: NCBI. ClinVar [VCV000560389.1], https://www.ncbi.nlm.nih.gov/clinvar/variation/VCV000560389.1 (accessed 3 November 2020).
NM_001130089.1(*KARS1*): c.871T>G (p.Phe291Val) *GRCh37:* Chr16:75669586 *GRCh38:* Chr16:75635688	Pathogenic	Scheidecker S et al. [14]
NM_001130089.1(*KARS1*): c.517T>C (p.Tyr173His) *GRCh37:* Chr16:75670401 *GRCh38:* Chr16:75636503	Pathogenic	Santos-Cortez et al. [8] Szafranski et al. [10]
NM_005548.2(*KARS1*): c.430_431dup (p.Tyr145fs) *GRCh37:* Chr16:75670402-75670403 *GRCh38:* Chr16:75636504-75636505	Pathogenic	McLaughin HM et al. [15]
NM_005548.2(*KARS1*): c.314T>A (p.Leu105His) *GRCh37:* Chr16:75674156 *GRCh38:* Chr16:75640258	Pathogenic	McLaughin HM et al. [15]
NM_005548.2(*KARS1*): c.1493C > T (p.Ala498Val) *GRCh37:* Chr16:75663371 (Second Variant) *GRCh38:* Chr16:75629473	Pathogenic	Murray CR et al. [16] THIS PAPER (2020)
NM_005548.3(*KARS1*): c.1258C > T (p.Arg420Cys) *GRCh37:* Chr16:75665146 *GRCh38:* Chr16:75631248	Conflicting Interpretations	Clinical testing, NO phenotype correlation: NCBI. ClinVar; [VCV000226679.7], https://www.ncbi.nlm.nih.gov/clinvar/variation/VCV000226679.7 (accessed 10 November 2020).
NM_005548.3(*KARS1*): c.1178G > A (p.Arg393Gln) *GRCh37:* Chr16:75665388 *GRCh38:* Chr16:75631490	Conflicting Interpretations	Liu et al. [17]
NM_001130089.1(*KARS1*): c.566 + 8G > A *GRCh37:* Chr16:75670344 *GRCh38:* Chr16:75636446	Conflicting Interpretations	Clinical testing, NO phenotype correlation: NCBI. ClinVar; [VCV000320638.3], https://www.ncbi.nlm.nih.gov/clinvar/variation/VCV000320638.3 (accessed 10 November 2020).
NM_005548.2(*KARS1*): c.22G > T (p.Glu8Ter) *GRCh37:* Chr16:75681516 *GRCh38:* Chr16:75647618	Likely Pathogenic	Clinical testing, NO phenotype correlation: NCBI. ClinVar [VCV000225009.1], https://www.ncbi.nlm.nih.gov/clinvar/variation/VCV000225009.1 (accessed 10 November 2020).
NM_005548.2(*KARS1*): c.599C > T (p.Pro200Leu) *GRCh37:* Chr16:75669880 *GRCh38:* Chr16:75635982	Likely pathogenic	KD Farwell et al. [18]

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
