# Peer review of "Progressive Early-Onset Leukodystrophy Related to Biallelic Variants in the KARS Gene: The First Case Described in Latin America"

_genes, 2020, doi:10.3390/genes11121437_

Round 1

Reviewer 1 Report

This is the third time that I review this paper. The authors have now added a table summarizing the KARS-1 gene variants known and have added in the discussion a nice review of other published cases. While the case they present adds very little new information to the overall knowledge on this topic, the added review of what is known provides for informative reading for those interested in the topic of KARS-1 associated leukodystrophies. I would recommend adding one sentence in the abstract stating that, in addition to the case report, the paper reviews the literature for other cases of KARS1 associated leukodystrophy. This will like be the major draw for other to want to read the paper.

Reviewer 2 Report

1) Line 51: Authors should specify the filtering steps (Variant quality and coverage threshold, population frequencies, impact on AA sequence like missense, stop, splicing…etc) or cite a paper where the filtering procedure used in this study is described. For Exome Sequencing performed on a trio is unlikely that only KARS variants were in the final list of variants found after the filtering steps. Was gene prioritization performed on the final list of candidate variants to search for candidate genes associated with the disease?

2) Line 101: "reported IN gnomAD of 0.00003", IN is missing

2) Line 197: “In Silico” does not need the “ “, just in silico

3) Line 199: “in vivo” does not need the “ “, just in vivo

Author Response

This manuscript is a resubmission of an earlier submission. The following is a list of the peer review reports and author responses from that submission.

Round 1

Reviewer 1 Report

Materials and methods

Line 49: The authors should specify which Illumina platform was used.

Line 49: The authors should change “66 readings” with more appropriate “66x”, in addition the authors should indicate at which coverage the 92% of all coding exons were covered? 1x or 10x or others threshold?

I suggest to move these two info (depth of coverage and percentage of exons covered) in the results section.

The authors should briefly describe how the total variants have been filtered for identifying the final candidate variants in KARS gene.

Results

Line 70: Typo “command. the child”

Line 90: KARS should be written in italic “KARS gene”. It should be also useful for the readers to know the NM code of the KARS isoform used.

Line 91: References citations in all the text are included in [] and not in ().

Line 92: The variant p.Ala526Val is recorded in gnomAD database, found in 1 allele over 15430 in a cohort of European (non-finnish) individual and it has been already recorded in dbSNP as rs1415687857. For both variants it should be useful to indicate they have extremely low frequency in public databases (<0.01%).

Did the authors validated the KARS variants by Sanger sequencing?

Did they check by Sanger sequencing the segregation of the KARS variants in the family? At least in the proband’s parents (Gi1: #2 and Gi1: #5)?

Discussion

Line 156: the sentence is not clear and contains typos, please rephrase it in a more comprehensible way.

Line 179: Was the WES performed on the trio or just on the proband (one sample)? The sentence seems to suggest that it was performed on parents and on proband, but in the methods it is stated that WES was performed only on the patient.

Line 190: “The use of WAS” should be corrected in “The use of WES”

Line 194: “In Silico” should be in italic “in silico

Line 196: “in vivo” should be in italic “in vivo

Line 198: the p.Ala526 is recorded in gnomAD and in dbSNP as rs1415687857

As in the results, a brief sentence about the segregation of the variants in the parents and in the other member of the family (if it was analyzed) should be added also in the Discussion.

Conclusions

Line 211: “…first case of a patient with a KARS mutation” should be changed in “…first case of a patient with KARS mutations”

Reviewer 2 Report

In the paper “Progressive early onset leukodystrophy related to 2 biallelic variants in the KARS gene: The first case 3 described in Latin America.” Vargas et all present a case report of a child with compound heterozygote pathogenic changes in the KARS gene with a resultant leukodystrophy. The paper is fairly well written, although it could benefit from editing for some sentence and grammatical structure that is awkward at times.

Major criticism:

  1. There are really only two novel findings presented in this paper: 1. This is the first case of a KARS associated leukoencephalopathy in Latin America, 2. This is the first paper to report one of the two KARS pathogenic variants observed in this patient.
  2. The compound heterozygote variants include a previously described variant and a second probably pathogenic variant not previously reported. This second variant is of interest as it results from an alanine to valine switch at position 526. This is quite interesting as an alanine to valine switch is a very conservative missense mutation (one small hydrophobic amino acid for another). The probable pathogenicity of this variant is briefly reviewed in the discussion, but the rather conservative nature of the missense mutation is never addressed.
  3. 371G > A (p.Ala526Val) cannot be correct. The cDNA variant is much more likely to be closer to position 1578 than 371. Please correct or explain what I am missing.
  4. The images do not contribute to illustrate the text very well (see below). They will need significant editing.
  5. It is not clear to me that figure 1, the pedigree, adds much to the paper.

Minor criticism:

Please note that proper nomenclature of genes is to italicize them.

Recommendation:

Consider writing a review of all previously reported cases of KARS pathogenic variants adding your patient as an index case. You could then include a figure illustrating the catalytic site of LysRS enzyme and a table illustrating previously described cases.

Figures/Images:

The figures are very problematic. It looks like whoever prepared the figures was somewhat confused as to what they were looking for. They appear to have extracted text directly from the radiology report and picked images that somewhat vaguely related to the report. They did not label any of the figures. As a result, the figures are extremely hard to make any sense off.

Minor point: The use of the terminology figures and images is rather unusual. Most people would label all figures and images simply as figures.

Image 1. None of the images are labeled and the legend does not appear to correspond to is evident in the images. There are areas of hyperintensity in the right greater than left occipital cortex and cerebellar hemispheres. It is not clear what the T1 sequence adds. On the other hand, it is difficult to understand what the authors describe when they point to the “fluid-attenuated inversion recovery (FLAIR) sequence, at the level of the middle cerebellar peduncles in the pons, lateral to medial lemniscus, there is a linear hyperintensity area, which looks hypointense in the T1 spin eco image.” The authors also state that there are no changes in the ADC and DWI sequences. This would be fine in the body of the text but does not belong in the legend since it is not shown.

Image 2. The legend does not illustrate the figure. The figure shows a T2 sequence: 2A-C, FLAIR: 2D-F, I think a T1 post contrast G-I and a sagittal T1 not labelled at all, a DWI: 2J-L and an ADC:2M-P. I cannot find the increased signal in the perirolandic area.

Minor point: Note the type: Perirrolandic should be spelled perirolandic.

Image 3. The same issues as for figure 1. Most of what is described in the legend is not obvious in the images. None of the images are labelled. The lactate peak is not labelled in the spectroscopy. On the other hand there appears to be a new arachnoid cyst in the frontal region that is not mentioned, neither is anything said about the very large signal artifact.

Round 2

Reviewer 2 Report

While the authors made significant changes to the paper, eliminated several figures and clarified the remaining figure, they did not address at all my major concern, that there is very little novelty in this paper. There are really only two novel findings presented in this paper: This is the first case of a KARS associated leukoencephalopathy in Latin America and this is the first paper to report one of the two KARS pathogenic variants observed in this patient. I had offered a suggestion to rescue the paper by turning it into of review of previously reported cases of KARS pathogenic variants adding their patient as an index case. Unfortunately, the authors have chosen not to do so and to resubmit the paper as case report. As written the paper adds very little of value to the available literature.

Please note the misspelling of leukodystrophy in the Keywords (leucodistrophy) and in the text, line 266 in the conclusion (leukodistrophy).